# Comparative Pharmacokinetic of Curcuminoids Formulations with an Omega-3 Fatty Acids Monoglyceride Carrier: A Randomized Cross-Over Triple-Blind Study

**DOI:** 10.3390/nu14245347

**Published:** 2022-12-16

**Authors:** Ester Cisneros Aguilera, Annick Vachon, Mélanie Plourde

**Affiliations:** Département de Médecine/Service de Gériatrie, Faculté de Médecine et des Sciences de la Santé, Université de Sherbrooke, Centre de Recherche sur le Vieillissement, Sherbrooke, QC J1H 4C4, Canada

**Keywords:** curcuminoids, curcumin, monoacylglycerol omega-3, absorption, pharmacokinetics

## Abstract

There is a growing interest for curcuminoids in the general population and the scientific research community. Curcuminoids, derived from turmeric spice, are lipophiles and therefore have a low solubility in water which hence have a low bioavailability in the human plasma. To circumvent this issue, a natural product developed by Biodroga Nutraceuticals combined curcuminoids with omega-3 fatty acids (OM3) esterified in monoglycerides (MAG). The objective was to perform a 24 h pharmacokinetics in humans receiving a single dose of curcuminoid formulated by three different means, and to compare their plasma curcuminoids concentration. Sixteen males and fifteen females tested three formulations: 400 mg of curcuminoids powder extract, 400 mg of curcuminoids in rice oil and 400 mg of curcuminoids with 1 g MAG-OM3. Blood samples were collected at 0, 1, 2, 3, 4, 5, 6, 8, 10 and 24 h post dose intake. Plasma samples were analyzed by ultra high-performance liquid chromatography with a triple quadrupole mass spectrometer (UPLC-MS/MS). Twenty-four hours after a single dose intake, the total plasma curcuminoids area under the curve (AUC) reached 166.8 ± 17.8 ng/mL*h, 134.0 ± 12.7 ng/mL*h and 163.1 ± 15.3 ng/mL*h when curcuminoids were provided with MAG-OM3, with rice oil or in powder, respectively. The C_max_ of total curcuminoids reached between 11.9–17.7 ng/mL at around 4 h (T_max_). One-hour post-dose, the curcuminoids plasma concentration was 40% higher in participants consuming the MAG-OM3 compared to the other formulations. Thus, in a young population, plasma curcuminoids 24 h pharmacokinetics and its increase shortly after the single dose intake were higher when provided with MAG-OM3 than rice oil.

## 1. Introduction

Enthusiasm and interest towards the consumption of curcuminoids is growing in the general population and in the scientific community. Curcuminoids are natural polyphenols extract representing 1–6% of the commonly known turmeric spice (Curcuma longa) and are composed of curcumin (60–77%), demethoxycurcumin (DMC) (17–27%) and bisdemethoxycurcumin (BDMC) (3–15%) [1,2,3]. Curcumin is a natural bioactive molecule that is recognized, in vitro, for its anti-inflammatory [4], antibacterial, antiviral, antifungal [5] and antioxidant [6] functions.

One major limitation with the use of curcuminoids is that they are lipophiles, hence they have a low oral absorption and plasma bioavailability even when provided at dosages of 1000 mg [7,8,9,10,11]. To circumvent this issue, many alternatives have been designed to make them more soluble in water and hence improve their bioavailability [12]. For instance, liposomes, nanoparticles, phospholipids, piperine were used as a carrier for curcuminoids [5,11,13]. Although most of these alternative improved somehow curcuminoids plasma bioavailability, some, like piperine, has been reported to inhibit CYP3A4 drug metabolizing enzyme and could lead to potential toxicity with specific drugs such as lovastatin, ketoconazole, triazolam, etc. [14,15]. Hence, using a combination of natural health product to improve curcuminoids plasma concentrations is of great interest in the field to stay as much as possible free of synthetic products to increase curcuminoids plasma bioavailability.

In a previous pharmacokinetic study conducted by our group, we compared plasma concentrations of omega-3 fatty acids (OM3) esterified in monoacylglycerol (MAG), ethyl esters or triglycerides [16]. The study included young males and females (n = 11/sex) and tested a single oral dose of 1.2 g of OM3 esterified in either of the three forms. Plasma OM3 were 2–3 times higher when provided in MAG-OM3 than when esterified in ethyl esters or triglycerides [16]. MAGs have a glycerol that is hydrophilic and a fatty acid that is hydrophobic which hence make it amphipathic, a characteristic that can improve curcuminoids emulsification [17,18].

Therefore, to take advantage of the amphipathic characteristics of MAG-OM3, we hypothesized that curcuminoids with MAG-OM3s will, over a 24 h pharmacokinetics, increase plasma curcuminoids more than when provided formulated in powder or with rice oil, a neutral oil.

The objective of this study was to compare the plasma concentration of curcuminoids over 24 h when it was provided as a single oral dose with MAG-OM3s, rice oil (neutral oil) or provided in powder.

## 2. Materials and Methods

### 2.1. Study Participants

The study was a triple-blind, randomized, crossover clinical trial and it was conducted at the Research Center on Aging, Centre Intégré Universitaire de Santé et des Services Sociaux. The Research Ethics Committee of the CIUSSSE-CHUS approved the study protocol (reference no 2020-3569). This study is registered on https://clinicaltrials.gov with the registration number: NCT04382014 and could be accessed 5 June 2020.

Recruitment was conducted between July 2021 and January 2022. Inclusion criteria were males and females between 18 and 50 years old; not taking any curcuminoids or OM3 supplements 6 months prior the study and during the study; have a body mass index (BMI) between 18.5 and 34.9 kg/m^2^; have normal to moderately high lipidemia and using a contraceptive method during the study for women of childbearing age.

Exclusion criteria were allergy to fish or seafood; under a specific diet like vegetarian or vegan; performing more than 180 min/week of intense physical activity or more than 210 min/week of moderate intensity, having an history of current or past alcohol or drug abuse; smoking tobacco or marijuana; being or been an elite athlete; suffering from malnutrition; taking any anticoagulant or medication affecting fat absorption and lipid metabolism; taking any medication interacting with curcumin such as angiotensin II blockers, beta-blockers, calcium channels blockers, etc. Moreover, pregnant, or breastfeeding women; women with menopause or pre-menopause; anyone with a systemic disease, or having a liver, kidney or thyroid disease were excluded. Other exclusion criteria include having a cardiac event in within 6 months prior starting the study; having an history of thrombosis or hemorrhagic diathesis; having a gastrointestinal disease, a neurodegenerative disease, a genetic disorder, a psychiatric history, or a neurological disorder; having any major surgery in the last year or having donated blood in the last 30 days.

Participants corresponding to these inclusion/exclusion criteria were thereafter invited for an on-site screening visit. They had to be fasted for 12 h prior the screening visits and not consume alcohol 48 h prior the visit. All participants provided informed written consent before any procedure was conducted. After obtaining their written consent, the following measures were collected: weight, height, blood pressure, pulse, and a fasted blood sample.

Fasting glucose, total cholesterol, triglycerides, cholesterol high density lipoproteins (HDL-C), cholesterol low density lipoproteins (LDL-C) and glycated hemoglobin (HbA1c) were quantified in the fasted blood samples. Blood biochemistry was performed at the Centre Hospitalier Universitaire de Sherbrooke clinical laboratory.

### 2.2. Study Products

The participants tested three different supplements produced by Biodroga Nutraceuticals: curcuminoids extract emulsified in 760 mg of an OM3 oil esterified in MAG providing 350 mg of eicosapentaenoic acid and 152 mg docosahexaenoic acid (NPN 80107639), curcuminoids extract combined with rice oil (NPN 80102058) and curcuminoids extract in powder with 23% microcrystalline cellulose and 0.2% maltodextrin (NPN 80102057). Each capsule contained 200 mg of curcuminoids extract. Curcuminoids single dose supplementation in humans is considered safe and well tolerated without any serious adverse effects [7].

Curcuminoids content of capsules containing only powder, rice oil or MAG-OM3 were analyzed using the whole weighted content. Briefly, for powder capsules, whole weighted content was added to 25 mL of reagent alcohol, vortexed for 30 s and sonicated for 30 min. For capsules containing rice or MAG-OM3, 25 mg of the capsule content was added to 25 mL of reagent alcohol, vortexed for 30 s and sonicated for 30 min. The mixtures from both extraction procedure was thereafter similar. Mixtures remained at room temperature for 5 min, 10 µL was added to 50 mL of methanol and vortexed for 15 s. 1 mL of the supernatant was filtered using 0.22 µm filter made of polytetrafluoroethylene. One hundred µL of the solution was transferred with 150 µL of internal standard composed of Curcumin-d6 as mentioned in the extraction from plasma, vortexed and 200 µL of the solution was transferred into vials for analysis. Quantification was made by ultra high-performance liquid chromatography with a triple quadrupole mass spectrometer (UPLC-MS/MS) as described below. The fatty acid profile of capsules was also performed as previously described Chevalier et al. [19] Curcuminoids levels and fatty acid profiles are provided in Table 1 and Table 2.

### 2.3. Study Design

The study design is depicted in Figure 1. Participants tested the three formulations of curcuminoids in a random order with a minimum of 7-day washout period between treatments. Randomization was conducted using https://www.randomizer.org/. Participants, research nurse and experimenter were blinded to supplement allocation. On each study visit, participants were fasted for 8 h prior the visit and avoided drinking alcohol 24 h prior their appointment. A catheter was installed in the participant’s forearm by the research nurse. After collecting a first fasted blood sample of 5 mL (T0), participants were provided a low-fat standardized breakfast of 410 kcal (75% carbohydrates, 22% proteins, 3% lipids) composed of a banana, two toasts with jam, a protein milk and a coffee or a tea. Two capsules were orally administrated to the participants while they were taking their breakfast. The capsules were not all identical in shape and color, therefore participants were blindfolded to remain blinded. After the supplement intake, 5 mL of blood was collected in EDTA vacutainer tubes at the following timepoints: 1, 2, 3, 4, 5, 6, 8, 10, and 24 h (10 samples per treatment per participant). Samples were stored on ice before being centrifuged at 1700 *g* for 10 min at 4 °C. Plasmas were aliquoted and stored at −80 °C until analysis. Each plasma sample was coded with a random number to keep the personnel blinded to the participant’s number, its treatment and the timepoint at which the blood sample was collected. This procedure enables to keep the personnel blinded when performing biochemical analysis thereafter.

During the pharmacokinetic day other meals such as a lunch (614 kcal, 78% carbohydrates, 17% proteins, 5% lipids) composed of one serving of chow mein, baby carrots, an apple, a yogurt, a juice and a fig bar, dinner (696 kcal, 72% carbohydrates, 23% proteins, 5% lipids), composed of one serving of General Tao, one slice of bread, one mini cucumber, one apple sauce, a protein-enriched milk, a tomato juice and a granola bar were provided. An evening snack (210 kcal, 87% carbohydrates, 6% proteins, 7% lipids) was composed of one granola bar and one package of gummy fruits was also provided at the end of the day and before participants leave the clinical room for the evening. Participant were instructed to refrain from eating other food in the evening and until a last blood sample collection the day after (T24 h). Menus and amount of food consumed for each participant was monitored and identical at each study day. Drinking water was recommended throughout the study days. Lunch was provided 4 h after the supplement intake whereas dinner and snack were provided 9 h and 10 h after the supplement intake.

### 2.4. Curcuminoids Extraction from the Plasma

Curcuminoids were extracted from plasma samples according to the method of Cuomo et al. [3] Briefly, the plasma samples were thawed for 30 min at room temperature. In the meantime, 50 µL of internal standard solution (250 ng/mL) composed of Curcumin-d6 (ISTD: CND Isotope, Pointe-Claire, QC, Canada) in methanol was added in an empty Eppendorf tube of 1.5 mL to which 0.2 mL of plasma and 100 µL of β-Glucuronidase/Arylsulfatase 10,000 U/mL from *Helix pomatia* (Sigma-Aldrich, Saint-Louis, MO, USA) diluted in phosphate buffer 0.1 M, pH = 6.86 were added. The tubes were vortexed for 1 min and incubated at 37 °C for 1 h to hydrolyze the phase-2 conjugates of curcuminoids [20,21]. Then, curcuminoids were extracted two times from the mixture by adding 1 mL of ethyl acetate, then vortexed 1 min and sonicated for 15 min. After sonification, tubes were centrifugated at 15,000 *g* at room temperature for 6 min. The superior organic phase was then transferred, and the process was repeated a second time. The combined organic phase was evaporated under nitrogen stream. The curcuminoids extract was reconstituted in 100 µL of methanol, centrifuge at 5000 *g* for 5 min to remove any particles, and 50 µL of the supernatant was then transferred to HPLC vials.

### 2.5. Curcuminoids Analysis

The samples were stabilised at 4 °C in the autosampler of the UPLC-MS/MS for 30 min prior its injection (4 µL). Calibration curve was made fresh every day of analysis, and quality controls were introduced at the beginning and at the end of the sequence. The calibration curve uses six concentrations ranging from 0 to 150 ng/mL made in a blank plasma matrix using a mixture of the three standard stock solutions of curcumin, BDMC and DMC (Sigma-Aldrich, Saint-Louis, USA). The low and high control quality were at concentration of 8 ng/mL and 200 ng/mL. Analyses of curcumin, BDMC and DMC levels were performed on a Waters Acquity UPLC H-Class system coupled to Waters Xevo TQ-S Micro tandem mass spectrometer with ESI source in positive mode (Waters, Milford, MA, USA). Curcuminoids were separated on a Waters Acquity UPLC BEH C18 column (2.1 × 50 mm, 1.7 µm) coupled with a Waters Acquity UPLC BEH C18 VanGuard pre-column (2.1 × 5 mm, 1.7 µm) at 40 °C. Mobile phase A was composed of Acetonitrile: H_2_0 (50:50, *v*/*v*) with 0.1% formic acid and mobile phase B was composed of Acetonitrile with 0.1% formic acid. The gradient elution program was as follows: 0–1.5 min (0–0% B), 1.5–1.6 min (0–95%B), 1.6–4.5 min (95–95% B), 4.5–4.6 min (95–0%B) and 4.6–6 min (0–0%B). The flow rate was 0.5 mL/min, and the total run time was 6 min

Nitrogen desolvation was set at 600 °C with a flow rate of 1000 L/h, the nitrogen cone gas was set at 80 L/h and the argon collision gas was set at 5 psi. The source temperature was at 150 °C and the capillary voltage was set at 1.0 kV. The cone voltage was 22 V for curcumin, 28 V for curcumin-d6, 34 V for DMC and 22 V for BDMC and the collision energy was at 22 eV. Quantification was performed in multiple reaction monitoring mode (MRM) with the following transition monitored: curcumin (*m*/*z* 369.1 → 177.1); DMC (339.1→ 147.0), BDMC (309.0 → 147.1) and curcumin-d6 (375.1 → 180.1) with a dwell time of 0.025 s. Mass Lynx software (Waters corporation) was used to generate the calibration curves by plotting the peak area ratio of curcumin, BDMC and DMC to internal standard (curcumin-d6). Each curcuminoid generated a calibration curve daily. Correlation coefficients for all calibration curves reached R^2^ > 0.99. The concentrations of curcuminoids of the plasma samples were calculated using a linear regression using the 1/x weighting factor.

### 2.6. Statistical Analysis

The primary outcomes of this study were area under the curve between 0 h and 24 h (AUC 0–24 h), between 0 h and 6 h (AUC 0–6 h), the maximal concentration (C_max_) and the time to reach it (T_max_). The AUC 0–24 h was used to calculate our sample size since it is more evaluated in the pharmacokinetic studies than AUC 0–6 h. Using the website https://clincalc.com/stats/samplesize.aspx [22] and data from a published study with a similar study design than this study [3], we used mean AUC 0–24 h ± SD from this study comparing Meriva (640.2 ± 197.7 ng/mL*h) vs. control (202.8 ± 53.8 ng/mL*h). Using an alpha = 0.05 and a power of 80%, we calculated that n = 5/group was required. Another similar study tested curcuminoids pharmacokinetics in 6 to 23 participants [13]. Here, we wanted to have enough statistical power to test men and women apart if required. Hence, we recruited 31 participants (16 males and 15 females) which gives us enough statistical power.

Statistical analyses were performed using GraphPad Prism version 9.3.1 for Windows (GraphPad Software, San Diego, CA, USA). The anthropometric characteristics were analyzed using an unpaired t test if data were normally distributed and if not, a Mann–Whitney non-parametric was used. *p* value < 0.05 were considered statistically significant. Two-way ANOVA for repeated measures was performed to assessed statistical difference between groups with time and treatment and the testing of their interaction. The independent variable was time (h), and the dependent variable was treatment (supplements) expressed in curcuminoids concentration (ng/mL). Multiple comparisons were performed with the Tukey test if the ANOVA result was significant. The AUC, from 0 h to 24 h, and 0 h to 6 h were calculated using the trapezoid rule and generated the C_max_ and T_max_ in GraphPad. AUC, C_max_, and T_max_ were not normally distributed, and the Friedman test was used to compare groups. If significant, then the Dunn’s test was used for multiple comparison. To account for multiple testing, results were considered significant when *p* < 0.025 in the post hoc analysis. To compare results from males and females, a two-way ANOVA was performed to compare treatments, AUC, C_max_ and T_max_. The independent variable was sex, and the dependent variables were AUC, C_max_ and T_max_.

## 3. Results

Seventy-eight persons were contacted or manifested their interest to participate to the study (Figure 2). They were screened by telephone with an eligibility questionnaire, and thirty-six persons met the inclusion criteria afterward. Five participants were thereafter excluded: two had a vasovagal syncope during blood puncture and were advised to not continue the study, one had potassium levels above 5.1 mmol/L, one had cortisol level higher than 624 nmol/L, and one was eligible, but our recruitment was completed by the time we received blood results. Hence, thirty-one participants were included and completed the study (16 males and 15 females).

Participants’ anthropometric characteristics are described in Table 3. The mean age of participants was 27.6 ± 6.0 years old, and the average body mass index was 25.1 ± 3.8 kg/m^2^. The results of blood profiles were all within clinical normal range. There were no statistical differences between males and females’ anthropometric characteristics.

### 3.1. Primary Outcomes

Pharmacokinetics parameters were assessed in males and females and there was no statistical difference between male and female supporting why the primary outcome was detailed with the whole cohort. The primary outcomes were to determine the AUC (0–24 h), AUC (0–6 h), C_max_ and T_max_ pharmacokinetics parameters for total curcuminoids upon receiving the three curcuminoids formulations. Total curcuminoids 0–24 h AUC was 24% higher when provided with MAG-OM3 compared to rice oil (*p* = 0.0297) (Figure 3B). The 0–6 h AUC was 42% higher when curcuminoids were provided with MAG-OM3 compared to rice oil formulation (*p* = 0.0230) (Table 4). The maximum concentration of total plasma curcuminoids range between 11.9–17.7 ng/mL and did not differ between the three formulations. This maximal concentration was reached approximately 4 h after the single dose intake regardless of the formulation.

### 3.2. Secondary Outcomes

Pharmacokinetics parameters of secondary outcomes are described in Table 4. Time required to reach the maximum concentration in curcumin, DMC and BDMC were around 5 h, 4 h, and 1 h, respectively for all formulations (Table 4). The maximum concentrations (Table 4) were between 9.1–12.9 ng/mL for curcumin, 2.0–3.3 ng/mL for DMC and 1.5–2.3 ng/mL for BDMC.

Plasma concentration of total curcuminoids, curcumin, DMC and BDMC 1 h after the single dose intake were significantly higher when provided with MAG-OM3 compared to when they were provided in powder (Figure 4). Twenty-four hours after the single dose intake, plasma concentrations in curcuminoids were under 2.52 ng/mL no matter how they were formulated.

Using a non-parametric Friedman test, 0–24 h AUC of curcumin, DMC and BDMC were not significantly different no matter how they were formulated (Table 4). The 0–6 h AUC of curcumin (*p* = 0.0337), DMC (*p* = 0.0113) and BDMC (*p* = 0.0063) were at least 40% higher when provided with MAG-OM3 compared to the rice oil formulation (Table 4).

## 4. Discussion

In this study, we hypothesize that curcuminoids provided with MAG-OM3s will increase plasma curcuminoids more than when provided formulated in powder or with rice oil. Before testing our primary outcome with the whole cohort, we investigated whether male’s pharmacokinetics was different from that of female’s. Our data support that there was no significant sex by supplement interaction for any outcomes and for any formulations supporting that data from the whole cohort (male + female) can be analyzed together.

Twenty-four hours after the single dose intake of curcuminoids with MAG-OM3, rice oil and powder, the total curcuminoids in the plasma 0–24 h and 0–6 h AUC were higher when provided with MAG-OM3 compared to when provided with rice oil although C_max_ and T_max_ were not different between treatments. Hence, our hypothesis is partially rejected. After the 24 h follow-up, plasma curcuminoids returned close to baseline no matter the supplement. In previous pharmacokinetic studies evaluating different formulations, the 0–24 h AUC range between 95.3 ± 4.6 ng/mL*h and 4474.7 ± 1675.2 ng/mL*h [13,23,24] and the 0–6 h AUC available range between 80 ± 10 ng/mL*h and 274.35 ± 169.57 ng/mL*h [25,26] which are ranges similar to those reached in this study. However, differences between previous studies and the one performed here can be explained by study design, number, and selection criteria of participants, curcuminoids dosages, etc. Curcuminoids with MAG-OM3 had a similar plasma-curcuminoid response compared to LongVida [27] a formulation made of phosphatidylcholine with a dose of 650 mg reaching a C_max_ of 22.4 ± 1.9 ng/mL and a T_max_ of 2.4 ± 0.4 h vs. To MAG-OM3 (C_max_ = 17.7 ± 2.3 ng/mL, T_max_ = 3.4 ± 0.5 h). Additionally, micronized curcumin [23] had a comparable response than MAG-OM3 curcuminoids formulation with a C_max_ of 15.3 ± 8.9 ng/mL and a T_max_ of 8.8 ± 6.4 h. Nonetheless, other alternatives with superior plasma concentrations are also available like Meriva [3] using phytosomes with a curcuminoid dose of 376 mg reaching a C_max_ = 209.9 ± 54.9 ng/mL and a T_max_ of 3.8 ± 0.6 h and NovaSol [23] (410 mg of curcuminoids) using micelles with 93% tween 80 reaching a C_max_ of 1189.1 ± 518.7 ng/mL and a T_max_ of 1.1 ± 0.4 h. However, these alternatives are mostly synthetics, and some have potential adverse effect such as potential implication in renal and liver toxicity caused by tween which should be carefully considered [28,29,30].

This study also generated an unexpected result with the curcuminoid powder since its 0–24 h AUC plasma curcuminoids was high compared to powders used in other studies even when provided at high doses [7,31]. Here, the powder contained very low concentrations of total lipids (Table 2) but contained 23% of microcrystalline cellulose [32,33,34] which is an emulsifier and a fat substitute and 0.2% of maltodextrin [35] known to enhance solubility. The use of these excipients probably explains the high 0–24 h AUC of the powder and the curve shape where the increase in curcuminoids was delayed in time compared to other treatments containing oil although reaching a similar peak curcuminoid concentration in the plasma. In other studies where curcuminoid powder not containing these excipients were used, C_max_ was to nondetectable to 9.0 ± 2.8 ng/mL with doses ranging from 30 mg to 2000 mg [13]. Here, 400 mg of the curcuminoid powder generated, a C_max_ = 15.7 ± 1.6 ng/mL suggesting that the excipients in the formulation might improve curcuminoids bioavailability.

Our result also supports that one hour after the single dose intake, curcumin (*p* < 0.0001), BDMC (*p* < 0.0001), DMC (*p* < 0.0001) and total curcuminoids (*p* < 0.0001) plasma concentration levels were significantly higher when consumed with MAG-OM3 compared to other formulations. Although, C_max_ of curcumin, BDMC, DMC and total curcuminoids did not differed between treatments, the timing of these concentrations differed between treatments. This result support that MAG-OM3 are good carriers of curcuminoids in the early phase of the postprandial transit, hence delivering these compounds at a faster rate than other formulation. MAGs are directly absorbed by enterocytes without needing to be hydrolyzed by pancreatic lipases [36,37] which leads to a predigested form compared to triglycerides and therefore facilitate the absorption and lowers risk of any gastrointestinal side effects [38,39]. For patients with malabsorption disorder, this could represent a significant benefit [40]. Therefore, with a faster gastrointestinal absorption those with an inflammatory bowel disease (IBD) such as ulcerative colitis and Crohn’s disease could reach high levels of curcuminoids in the colon tissue and with a potential benefit on reducing inflammation [41]. To lower inflammation, on study suggested that curcumin and omega-3s could be part on a nutritional intervention for IBD patients [42]. Indeed, OM3 could increase the potential anti-inflammatory response of curcuminoids since both are cyclooxygenase-2 (COX-2) inhibitors. COX-2 is known as a key enzyme in the conversion of arachidonic acid to prostaglandins which are pro-inflammatory mediators [43,44,45]. OM3 such as EPA and DHA are also implicated in the arachidonic acid pathway as direct competitor for the synthesis of anti-inflammatory mediators by the COX-2 pathway [46]. However, this hypothesis will need to be further tested in an independent study. In this study, we also observed a biphasic shape of the curcuminoids pharmacokinetic when they were provided with oil. One main explanation for this biphasic distribution is related to curcuminoids redistribution from tissues and their enterohepatic recycling after food intake 4 h and 9 h post-single dose intake [47,48]. Hence, when ingested with oil, curcuminoids were more rapidly secreted in the plasma after breakfast whereas when provided in powder, which was consumed with a low-fat breakfast, it took longer to curcuminoids to reach blood circulation. Thus, we hypothesize that curcuminoids, like CoQ10 which is also liposoluble, might have been stocked in the liver or other tissues enriched in lipids and thereafter redistributed to the blood circulation after food intake hence explaining a biphasic shape of the pharmacokinetics curve [49,50].

The present study had strengths and limitations. The cross-over randomized design was a strength that allows to minimize the interindividual variability between treatment since the same participant tested the three formulations of curcuminoids and was therefore its own control. This study design also reduces the number of participants required to have enough statistical power to detect significant differences between the groups and minimize type 1 error. Males (n = 16) and females (n = 15) were balanced whereas most other studies were done only on males [27,51,52] or included only one female (8 males: 1 female) [3,31]. Standardized procedures, with respect to meals and snacks and the intake of the same food from one treatment to another were also strengths. Our menus, however, provided low fat meals to minimize how fat in the meals can modulate absorption and bioavailability of the curcuminoids. To avoid any experimental and selection bias, treatment allocation was randomized, and the design was triple blinded which meant that neither the participants, the research nurse and the experimenter knew which treatment was given and in what order. All sample collected were coded with a random number and blinding was kept until all biochemistry analyses were terminated.

In terms of limitations, this pharmacokinetic study tested only one dose of the products and nor efficacy, nor their pharmacodynamics were tested, hence limiting the extend to which these results can be transferred to the general population. This study was also limited to healthy males and females between 18 and 50 years old without severe health conditions. Additionally, the standardized low-fat meal is not representative of the usual North American diet and limit transfer of these data to the general population.

## 5. Conclusions

In conclusion, participants consuming curcuminoids with MAG-OM3 had higher plasma curcuminoids concentration over 24 h compared to when curcuminoids was provided with rice oil. Moreover, in postprandial, especially 1 h after the single dose intake, curcuminoids with MAG-OM3 increased more curcuminoids plasma concentrations compared to other formulations supporting that curcuminoids were available faster than the other formulations.

## Figures and Tables

**Figure 1 nutrients-14-05347-f001:**
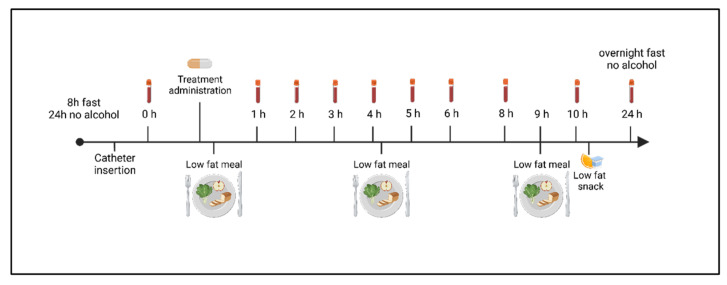
Timeline of the study procedure for a treatment visit followed by a 7-day washout.

**Figure 2 nutrients-14-05347-f002:**
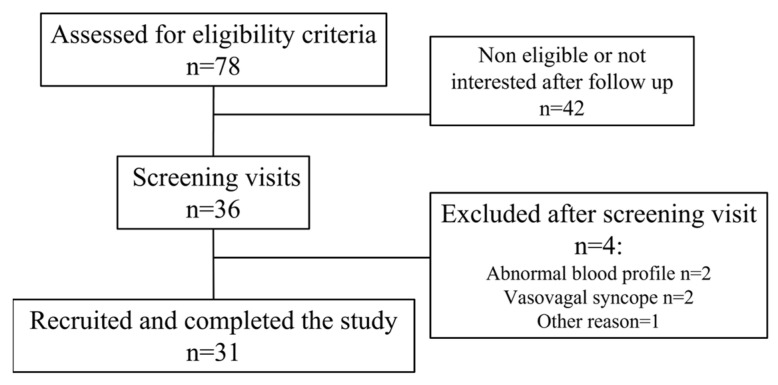
Flow chart of the study.

**Figure 3 nutrients-14-05347-f003:**
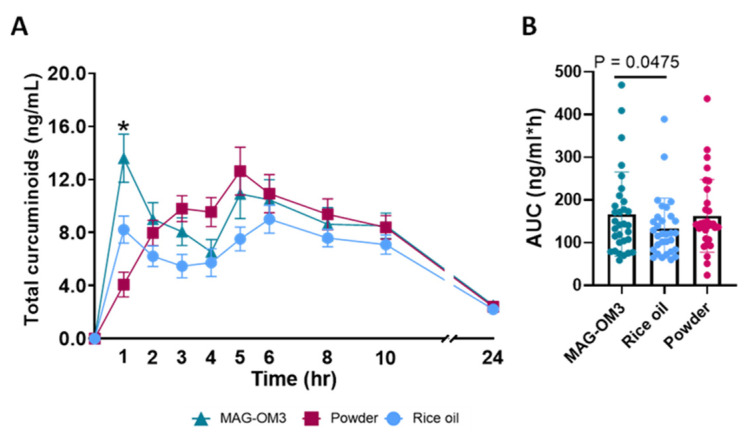
(**A**) Pharmacokinetic over 24 h of total plasma curcuminoids (ng/mL) after providing a single dose intake of curcuminoids combined with monoglycerides omega-3 fatty acids (MAG-OM3), rice oil or alone in powder. The data are expressed means ± SEM and are delta over the baseline. A repeated measure two-way ANOVA was performed to compare the different formulations and a Turkey’s multiple comparisons test was performed when the ANOVA was significant to determined which group different from the other. * Represent *p* < 0.05. (**B**) Area under the curve of total curcuminoids for each formulation (ng/mL*h). Data were compared using a Friedman non-parametric test followed by Dunn’s multiple comparison test when require.

**Figure 4 nutrients-14-05347-f004:**
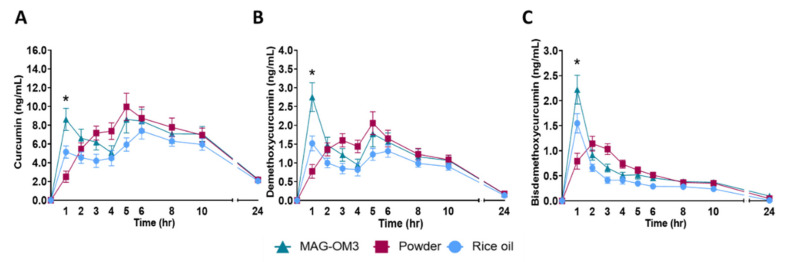
Pharmacokinetic over 24 h of total plasma (**A**) curcumin, (**B**) demethoxycurcumin and (**C**) bisdemethoxycurcumin (ng/mL) after providing a single dose intake of curcuminoids combined with monoglycerides omega-3 fatty acids (MAG-OM3), rice oil or alone in powder. The data are expressed means ± SEM and are delta over the baseline. A repeated measure two-way ANOVA was performed to compare the different formulations and a Turkey’s multiple comparisons test was performed when the ANOVA was significant to determined which group different from the other. * Represent *p* < 0.05.

**Table 1 nutrients-14-05347-t001:** Curcuminoids content in the supplements ^1^.

	Curcuminoids + MAG-OM3 Supplement (mg/Capsule)		Curcuminoids + Rice Oil Supplement (mg/Capsule)		Curcuminoids Powder Supplement (mg/Capsule)		
Curcuminoids	Mean	SD	Mean	SD	Mean	SD	*p*-Value
Curcumin	147.6	3.4	148.6	8.2	128.8	5.8	0.0839
Bisdemethoxycurcumin (BDMC)	5.8	0.3	5.7	0.6	5.7	0.4	0.9945
Desmethoxycurcumin (DMC)	34.7	0.7	34.9	2.2	31.5	1.6	0.2044
Total Curcuminoids	188.0	4.3	189.2	11.0	166.1	3.9	0.3287

^1^ n = 6 capsules/type of supplement, MAG-OM3 = Omega-3 esterified in monoglycerides. Curcuminoids content of type of supplements were compared using a one-way ANOVA.

**Table 2 nutrients-14-05347-t002:** Fatty acids profile of the supplements ^1^.

	Curcuminoids + MAG-OM3 Supplement (mg/Capsule)		Curcuminoids + Rice Oil Supplement (mg/Capsule)		Curcuminoids Powder Supplement (mg/Capsule)	
Fatty Acids	Mean	SD	Mean	SD	Mean	SD
C8:0	0.4	0.7	0.5	0.8	0.0	0.0
C10:0	1.0	1.0	0.6	1.1	0.0	0.0
C14:0	2.5	0.2	2.8	0.1	0.0	0.0
C16:0	18.9	0.6	161.9	4.6	5.9	0.3
C16:1 n-7	1.8	0.2	1.5	0.1	0.0	0.0
C17:1	0.0	0.0	0.0	0.0	0.0	0.0
C18:0	12.9	1.0	17.9	0.5	4.3	0.2
C18:1 n-9	24.5	1.6	327.3	14.5	0.0	0.0
C18:1 n-7	5.6	0.5	7.1	0.3	0.0	0.0
C18:2 n-6	13.8	11.1	259.2	11.0	2.5	0.1
C18:3 n-6	1.1	1.0	2.7	0.4	0.0	0.0
C18:3 n-3	2.1	0.2	7.3	0.3	0.0	0.0
C20:0	3.0	0.2	6.4	0.3	0.0	0.0
C20:1	14.4	1.3	4.0	0.2	0.0	0.0
C20:2	0.0	0.0	0.0	0.0	0.0	0.0
C20:3 n-6	1.2	2.0	0.0	0.0	0.0	0.0
C20:4 n-6	19.8	2.0	0.0	0.0	0.0	0.0
C20:3 n-3	0.0	0.0	0.0	0.0	0.0	0.0
C20:5 n-3	372.9	36.6	0.0	0.0	0.0	0.0
C22:0	0.0	0.0	2.4	0.1	0.0	0.0
C22:1	0.0	0.0	0.0	0.0	0.0	0.0
C24:0	0.0	0.0	0.0	0.0	0.0	0.0
C22:5 n-6	0.0	0.0	0.0	0.0	0.0	0.0
C22:5 n-3	26.3	2.7	0.0	0.0	0.0	0.0
C22:6 n-3	153.5	15.9	0.0	0.0	0.0	0.0
Total concentration	675.7	78.8	801.6	34.3	12.8	0.6

^1^ n = 3 capsules/type of supplement, MAG-OM3 = Omega-3 esterified in monoglyceride.

**Table 3 nutrients-14-05347-t003:** Anthropometric characteristics of the participants ^1^.

Anthropometric Characteristics	Total (n = 31)	Male (n = 16)	Female (n = 15)	*p*-Value
Age (years)	27.6 ± 6.0	28.9 ± 6.8	26.3 ± 5.0	0.2346
BMI (kg/m^2^)	25.1 ± 3.8	25.8 ± 3.0	24.4 ± 4.4	0.3290
Plasma TG (mmol/L)	0.87 ± 0.51	0.91 ± 0.49	0.84 ± 0.54	0.4062
Plasma HDL-C (mmol/L)	1.39 ± 0.34	1.39 ± 0.35	1.40 ± 0.33	0.9659
Plasma LDL-C (mmol/L)	2.58 ± 0.70	2.68 ± 0.77	2.48 ± 0.62	0.4462
Plasma glucose (mmol/L)	4.64 ± 0.54	4.81 ± 0.54	4.45 ± 0.50	0.0691
HbA1c (%)	5.08 ± 0.30	5.13 ± 0.32	5.03 ± 0.28	0.4017

^1^ Results are means ± SD. BMI = body mass index, TG = triglycerides, HDL-C = high density lipoprotein cholesterol, LDL-C = low density lipoprotein cholesterol, HbA1c = glycated hemoglobin. Males and females were compared using an unpaired t-test whereas for plasma TG, data were not normally distributed, and a Mann–Whitney non-parametric test was used to compared males and females.

**Table 4 nutrients-14-05347-t004:** Pharmacokinetics parameters of each curcuminoid when provided with a different formulation ^1^.

Curcuminoid	Formulation ^2^	T_max_ (h)	*p*-Value ^3^	C_max_ (ng/mL)	*p*-Value	C24 h (ng/mL)	*p*-Value	C1 h (ng/mL)	*p*-Value	AUC 0–24 h (ng/mL*h)	*p*-Value	AUC 0–6 h (ng/mL*h)	*p*-Value
**Curcumin**	**MAG-OM3**	4.2 ± 0.6	0.2502	12.9 ± 1.6	0.1152	2.3 ± 0.3	0.9675	8.6 ± 1.2 A	<0.0001	134.3 ± 14.3	0.0307	39.3 ± 4.4 A	0.0337
	**Rice oil**	5.1 ± 0.5		9.1 ± 0.9		2.1 ± 0.2		5.2 ± 0.7 A		110.1 ± 10.5		28.0 ± 3.3 B	
	**Powder**	5.8 ± 0.7		12.3 ± 1.3		2.2 ± 0.2		2.5 ± 0.6 B		132.0 ± 12.5		36.8 ± 4.1	
**Demethoxy-curcumin** **(DMC)**	**MAG-OM3**	2.3 ± 0.4 A	0.0009	3.3 ± 0.5 A	0.0102	0.2 ± 0.1	0.8440	2.8 ± 2.1 A	<0.0001	22.6 ± 2.7	0.0576	8.9 ± 1.0 A	0.0113
	**Rice oil**	3.5 ± 0.5 AB		2.0 ± 0.2 B		0.1 ± 0.1		1.5 ± 1.1 B		17.7 ± 1.8		6.1 ± 0.7 B	
	**Powder**	4.5 ± 0.4 B		2.5 ± 0.3 AB		0.2 ± 0.1		0.8 ± 1.0 B		22.2 ± 2.2		8.1 ± 0.9 AB	
**Bisdemethoxy-curcumin** (**BDMC)**	**MAG-OM3**	1.1 ± 0.1 A	<0.0001	2.3 ± 0.3 A	0.0066	0.10 ± 0.03	0.8465	2.2 ± 1.6 A	<0.0001	10.0 ± 1.0	0.1590	5.1 ± 0.5 A	0.0063
	**Rice oil**	1.1 ± 0.1 A		1.6 ± 0.2 B		0.01 ± 0.06		1.6 ± 1.1 B		7.1 ± 0.8		3.5 ± 0.4 B	
	**Powder**	1.8 ± 0.2 B		1.5 ± 0.1 B		0.05 ± 0.04		0.8 ± 0.9 B		9.3 ± 0.8		4.6 ± 0.4 AB	
**Total Curcuminoids**	**MAG-OM3**	3.4 ± 0.5	0.1681	17.7 ± 2.3	0.0939	2.5 ± 0.4	0.7283	13.6 ± 1.8 A	<0.0001	166.8± 17.8 A	0.0297	53.3 ± 5.9 A	0.0230
	**Rice oil**	4.3 ± 0.5		11.9 ± 1.1		2.2 ± 0.3		8.2 ± 1.0 B		134.0 ± 12.7 B		37.6 ± 4.3 B	
	**Powder**	4.8 ± 0.4		15.7 ± 1.6		2.4 ± 0.3		4.1 ± 0.9 C		163.1 ± 15.3 AB		49.5 ± 5.4 AB	

^1^ Results are mean ± SEM. ^2^ All formulations contain 400 mg of curcuminoid extract, MAG-OM3 = Omega-3 esterified in monoglycerides. ^3^ Data were not normally distributed, and a Friedman non-parametric test was used and if it met statistical difference to compare each formulation a Dunn’s multiple comparisons was used and statistical difference was represented by different letters.

## Data Availability

Not applicable.

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
