# Peer review of "Comparative Pharmacokinetic of Curcuminoids Formulations with an Omega-3 Fatty Acids Monoglyceride Carrier: A Randomized Cross-Over Triple-Blind Study"

_nutrients, 2022, doi:10.3390/nu14245347_

Round 1
Reviewer 1 Report
The objective of this manuscript was to perform a pharmacokinetics study in humans receiving a single dose of curcuminoid formulated by three different formula, and to compare their plasma curcuminoids concentration.
The the manuscript is well written. However, the conclusions seemed not be fully supported by the results. Therefore, I recommend that a minor revision. Here are few comments for you to consider.
Since the authors claimed the powder formulation containing 23% of microcrystalline cellulose and 0.2 % of maltodextrin, which might cause the enhancement of solubility for curcuminoids and further to improve their bioavailability. It is suggested to perform a solubility study for curcuminoids to further proof the concept.
Can the authors explain the reasons why there were two peaks phenomena in the pharmacokinetic profiles for MAG-OM3 and rice oil, but couldn’t be found in the powder group?
The authors claimed “participants consuming curcuminoids with MAG-OM3 had higher plasma curcuminoids concentration over 24 h compared to when curcuminoids was provided with rice oil.”
When comparing the PK parameters obtained from the three different formula, the only significant difference was the plasma concentration after 1 hour administration from each groups. However, there were no statistically significant difference when comparing the AUC, Cmax among these formulations. I would argue that curcuminoids with MAG-OM3 might only enhance the absorption rate of the curcuminoids.
Author Response
The manuscript is well written. However, the conclusions seemed not be fully supported by the results. Therefore, I recommend that a minor revision. Here are few comments for you to consider.
Answer: Thank you for the comment. Regarding the conclusion of the paper, we believe that it is well supported by the data since the area under the curve (AUC) is significantly higher in the MAG-OM3 group compared to rice oil only (See Figure 3B and Table 4). It is true that the MAG-OM3 formulation is not significantly higher that the powder, but our conclusion specifically mentioned that it is compared to the rice oil supplement. Hence, this was not modified in the revised version.
Since the authors claimed the powder formulation containing 23% of microcrystalline cellulose and 0.2 % of maltodextrin, which might cause the enhancement of solubility for curcuminoids and further to improve their bioavailability. It is suggested to perform a solubility study for curcuminoids to further proof the concept.
Answer: We thank that the reviewer for this suggestion. However, since we are in academics, we don’t have a Vessel Dissolution Apparatus to perform the specific solubility tests as suggested. Moreover, we provided four different references supporting this concept. Although we acknowledge that these references are not specific to the supplement used in this study, this is the only potential explanation we found for the high curcuminoids levels in the plasma after the intake of the powder supplement. Hence, the last sentence of this paragraph was tone down and now reads: ‘’ Here, 400 mg of the curcuminoid powder generated, a Cmax = 15.7 ± 1.6 ng/ml suggesting that the excipients in the formulation might improved curcuminoids bioavailability.’’
Can the authors explain the reasons why there were two peaks phenomena in the pharmacokinetic profiles for MAG-OM3 and rice oil, but couldn’t be found in the powder group?
Answer: Indeed, we observed a biphasic shape of the curcuminoids pharmacokinetic when they were provided with oil. One main explanation for this biphasic distribution is related to curcuminoids redistribution from tissues and their enterohepatic recycling after food intake 4h and 9h post-single dose intake (Bhagavan and Chopra 2007, Lambert and Parks 2012). Hence, when ingested with oil, curcuminoids were more rapidly secreted in the plasma after breakfast whereas the powder, which was consumed with a low-fat breakfast, took longer to reach blood circulation. Thus, we hypothesize that curcuminoids, like CoQ10 which is also liposoluble, might have been stocked in the liver or other tissues enriched in lipids and thereafter redistributed to the blood circulation after food intake (Pravst, Rodriguez Aguilera et al. 2020, Beaulieu, Vachon et al. 2022). This information was added in the discussion of the revised manuscript.
The authors claimed “participants consuming curcuminoids with MAG-OM3 had higher plasma curcuminoids concentration over 24 h compared to when curcuminoids was provided with rice oil.”
When comparing the PK parameters obtained from the three different formula, the only significant difference was the plasma concentration after 1 hour administration from each groups. However, there were no statistically significant difference when comparing the AUC, Cmax among these formulations. I would argue that curcuminoids with MAG-OM3 might only enhance the absorption rate of the curcuminoids.
Answer: In the primary outcome section 3.1, it is mentioned that the 24 h Area under the curve (AUC) is significant between the MAG-OM3 and the rice oil (Figure 3B (Ppost-hoc = 0.0475) and Table 4 (PANOVA = 0.0307)). There is indeed no significant difference in Cmax of curcumin but there are significant differences for DMC an BDMC (Table 4). Considering that the 24h AUC is significantly higher when curcuminoids were provided with MAG-OM3 vs with rice oil, we believe that our conclusion is correct and should not be modified.
References
Beaulieu, S., A. Vachon and M. Plourde (2022). "Women have higher levels of CoQ(10) than men when supplemented with a single dose of CoQ(10) with monoglycerides omega-3 or rice oil and followed for 48 h: a crossover randomised triple blind controlled study." J Nutr Sci 11: e2.
Bhagavan, H. N. and R. K. Chopra (2007). "Plasma coenzyme Q10 response to oral ingestion of coenzyme Q10 formulations." Mitochondrion 7 Suppl: S78-88.
Lambert, J. E. and E. J. Parks (2012). "Postprandial metabolism of meal triglyceride in humans." Biochim Biophys Acta 1821(5): 721-726.
Pravst, I., J. C. Rodriguez Aguilera, A. B. Cortes Rodriguez, J. Jazbar, I. Locatelli, H. Hristov and K. Zmitek (2020). "Comparative Bioavailability of Different Coenzyme Q10 Formulations in Healthy Elderly Individuals." Nutrients 12(3).
Reviewer 2 Report
This study has been conducted with appropriate design, method and statistics. Methods are explained well and clearly. Results are very clearly explained and tabulated.
Discussion part elaborated thoroughly considerimg all aspects. This will be very helpful for readers and researchers to think through further on the research in this area.
Overall, this research topic is of high interest and this study surely contributes with further insights on the use of cucurminoids with the MAG-OM3.
Author Response
This study has been conducted with appropriate design, method and statistics. Methods are explained well and clearly. Results are very clearly explained and tabulated.
Discussion part elaborated thoroughly considering all aspects. This will be very helpful for readers and researchers to think through further on the research in this area.
Overall, this research topic is of high interest and this study surely contributes with further insights on the use of cucurminoids with the MAG-OM3.
Answer: Thank you for these positive comments